# How to Ease the Pain of Taking a Diagnostic Point of Care Test to the Market: A Framework for Evidence Development

**DOI:** 10.3390/mi11030291

**Published:** 2020-03-10

**Authors:** Sara Graziadio, Amanda Winter, B. Clare Lendrem, Jana Suklan, William S. Jones, Samuel G. Urwin, Rachel A. O’Leary, Rachel Dickinson, Anna Halstead, Kasia Kurowska, Kile Green, Andrew Sims, A. John Simpson, H. Michael Power, A. Joy Allen

**Affiliations:** 1NIHR Newcastle In Vitro Diagnostics Co-operative, The Newcastle Upon Tyne Hospitals NHS Foundation Trust, Newcastle upon Tyne NE1 4LP, UK; sara.graziadio@newcastle.ac.uk (S.G.); amanda.winter@newcastle.ac.uk (A.W.); sam.urwin@newcastle.ac.uk (S.G.U.); rachel.oleary1@nhs.net (R.A.O.); rachel.dickinson2@newcastle.ac.uk (R.D.); andrew.sims5@nhs.net (A.S.); 2NIHR Newcastle In Vitro Diagnostics Co-operative, Room M2.088, Translational and Clinical Research Institute, William Leech Building, Medical School, Newcastle University, Newcastle NE2 4HH, UK; clare.lendrem@newcastle.ac.uk (B.C.L.); jana.suklan@newcastle.ac.uk (J.S.); will.jones@newcastle.ac.uk (W.S.J.); anna.halstead@newcastle.ac.uk (A.H.); Kasia.kurowska@newcastle.ac.uk (K.K.); kile.green@newcastle.ac.uk (K.G.); j.simpson@newcastle.ac.uk (A.J.S.); hmichaelpower@gmail.com (H.M.P.)

**Keywords:** point of care, medical device, diagnostic, care pathway analysis, value proposition, evidence generation, preparation for marketing, adoption, implementation

## Abstract

Bringing a diagnostic point of care test (POCT) to a healthcare market can be a painful experience as it requires the manufacturer to meet considerable technical, financial, managerial, and regulatory challenges. In this opinion article we propose a framework for developing the evidence needed to support product development, marketing, and adoption. We discuss each step in the evidence development pathway from the invention phase to the implementation of a new POCT in the healthcare system. We highlight the importance of articulating the value propositions and documenting the care pathway. We provide guidance on how to conduct care pathway analysis as little has been published on this. We summarize the clinical, economic and qualitative studies to be considered for developing evidence, and provide useful links to relevant software, on-line applications, websites, and give practical advice. We also provide advice on patient and public involvement and engagement (PPIE), and on product management. Our aim is to help device manufacturers to understand the concepts and terminology used in evaluation of *in vitro* diagnostics (IVDs) so that they can communicate effectively with evaluation methodologists, statisticians, and health economists. Manufacturers of medical tests and devices can use the proposed framework to plan their evidence development strategy in alignment with device development, applications for regulatory approval, and publication.

## 1. Introduction 

Bringing a diagnostic point of care test (POCT) to a healthcare market can be a painful experience as it requires the manufacturer to meet considerable technical, financial, managerial, and regulatory challenges. Scaling the ambitions to include marketing to more than one country brings additional complexity. Much pain can be avoided if the evidence needed to support decision-making around product development and marketing is planned, and then generated in stages in which progress and prospects are reviewed. This process is called stage-gating and aims to ensure that product development is cost and capital efficient. Our aim in this article is to help manufacturers of POCTs understand and implement this process, and to give them the concepts and vocabulary needed to discuss evidence development with statisticians and health economists. Manufacturers of *in vitro* diagnostics (IVDs) or other medical devices can readily adapt this framework to their own needs. Project management needs to be robust to ensure that the evidence development process remains under control and is delivered successfully.

We list the series of questions that a manufacturer might need to answer, and show graphically how this approach provides a high-level framework for planning the generation of evidence to support product development and ultimately marketing. Statistical and health economics methods are not described in detail. However, statisticians and health economists might well be interested in learning about the context in which their work is set.

### 1.1. An Example to Pin the Ideas On

We need an example of a medical device or diagnostic which requires evidence to be developed to guide investors and potential purchasers. To ensure that there is no confusion with living people, real companies, or actual health care systems, we will use a fictional scenario in which a device called NewTestR is proposed to be used to rule in or exclude a condition we will call ConditionR. ConditionR is usually managed in hospitals. NewTestR is more accurate than clinical diagnosis, and is provided as a point of care test that can be used in primary care by GPs or family physicians. The premium version of the device, called NewTestRx, is used to test for ConditionR and then provide advice to the patient on self-management of ConditionR at home. NewTestR would be considered for regulatory purposes to be an in vitro diagnostic medical device, IVD [1]. NewTestRx would be considered for regulatory purposes to be a medical device and a data driven, digital healthcare technology [2,3].

### 1.2. It’s a Bit More Complicated Than That

Our example is of a test that is intended to inform the diagnosis of a condition. However, tests may also be used to inform screening, monitoring, prognosis, and prediction for individuals of outcomes such as response to treatment [4]. 

Our example test provides a categorical result: positive or negative. However, many tests provide a quantitative result, often as a probability.

Our example is of a condition that is either present or absent. However, diseases are not dichotomous. They exist on a continuous spectrum, such as from mild to severe, from subclinical to obvious.

Our example is a single test for a single condition. However, it is becoming increasingly common for new devices to perform a panel of tests to inform the diagnosis or management of a single condition, or to inform the diagnosis of more than one condition.

Our example is of one test for a condition. However, tests are often used in a sequence determined by the result of a preceding test.

Although our example is a simplification of reality, it is the way that many tests are used and evaluated. Also, the use and evaluation of more complex testing is based on the straightforward case [5,6].

It is worth noting here that an item in the medical history or physical examination can be regarded as a diagnostic test and evaluated for accuracy using the same methods as a laboratory test.

## 2. A Framework to Guide Evidence Development for POCT IVDs and Other Medical Devices

From the outset of developing a new device a manufacturer will want to know what evidence they will need to ensure that the process is as efficient as possible [7]. Table 1 lists a series of questions that provides a framework for planning and managing product development. Figure 1 shows the evidence development pathway that follows from these questions. 

Although our framework quite closely reflects the requirements of the National Health Service (NHS) in England, it is straightforward to adapt it to other healthcare systems, provided differences in terminology are considered. The evidence required by national regulators, especially with respect to health economics assessments, will differ across jurisdictions. As even a summary of international variations in regulatory requirements would be outside the scope of this article, manufacturers should start familiarizing themselves with the relevant regulations early in their product’s journey along the evidence development pathway. 

The framework shown in Table 1 and Figure 1 was developed with IVDs in mind, but it is, as we will show, readily adapted to other types of medical devices and data-driven healthcare technologies.

The series of questions in Table 1 suggests that evidence should be developed along the pathway shown in Figure 1. We discuss each of the stages in the following sections. But first, we need to consider how to articulate the value propositions.

## 3. Articulating Value Propositions

NICE, the National Institute for Health and Care Excellence, recommend [8] that, when a company plans their market access strategy, they should:Understand the changing healthcare landscapeIdentify the most appropriate route to NHS accessExplore their value proposition with system stakeholders.

Companies may wish to obtain specialist help in doing this, whichever market they are considering. 

Our focus here is on how to articulate your value propositions, but this does not mean that understanding the healthcare landscape and routes to market access are unimportant. 

NICE say: “Achieving value for patients and the NHS is key to transforming healthcare. By engaging with system stakeholders early you can help ensure you are developing the new technologies and approaches that are needed” [8].

In contrast to NICE’s usage of “your value proposition”, singular and with a perspective focused on health economic measures such as cost-effectiveness, we would say “your value propositions”, plural, to draw attention to the fact that there is a range of stakeholders in the decision to adopt a new device or diagnostic, and that different stakeholders can have different perspectives, values, and priorities. The investor is likely to foreground financial returns, the purchaser and service provider the budget impact and cost effectiveness, and the patient and service provider the balance of potential clinical benefits and harms. 

The term “value proposition” has different meanings in business and in healthcare. Also, in healthcare it differs markedly between the US and the UK. We use “value propositions” (plural) with the ordinary language use to mean benefits, whether they are monetary or not.

The manufacturer will need to articulate a set of value propositions that will be attractive to all the stakeholders. Although the stakeholders will focus on different value propositions, they are, to use a sporting metaphor, all in the same game, and therefore each has an interest in the others’ priorities—assuming decision-making is well informed and rational. To ensure they have not overlooked a health value potentially provided by their device, manufacturers may find it helpful to consult the frameworks developed by Ferrante di Ruffano and Wurcel and their colleagues [9,10].
**Exercise: Value propositions**Write down the value propositions for the NewTestR and NewTestRx devices foregrounded by:InvestorsHealthcare fundersHealthcare provider organizations and their departmentsHealthcare provider professionalsPatients

If the exercise seemed difficult, or your answers felt incomplete, this may be because, while we have pointed out that articulating a comprehensive and coherent set of value propositions is essential, we have not yet discussed how this should be done. 

To articulate your set of value propositions, you will need to understand in some detail how your product will fit into and change care pathways, work and information flows. The tool for doing this is care pathway analysis.

## 4. Care Pathway Analysis—Some Necessary Theoretical Background

### 4.1. What is a Care Pathway?

The term “care pathway” is used in a variety of ways [11,12,13], including:The patient’s view of their journey through the healthcare system.The healthcare system’s view of the services provided, including workflows and information flows.The range of actual practices (for example, as documented by a clinical audit).The accepted best practice (for example as defined by clinical guidelines, standards, protocols).

We use “care pathway” to mean the healthcare system’s perspective of the services provided [4,6,14]. We are explicit when we discuss the patient’s view of their journey, clinical audit, and guidelines.

### 4.2. What is Care Pathway Analysis?

Care pathway analysis is the method of modelling a care pathway in a healthcare system. It is a type of systems or process analysis. The resulting model is shown graphically as a systems diagram or map of the services provided to a typical patient. Care pathway analysis is the starting point for assessing the cost and value of a healthcare service, and how this would change should a new technology be adopted. It is therefore worthwhile to consider how to represent a system graphically before looking at the process of analyzing a care pathway.

### 4.3. How Should a System be Represented Graphically?

A system has inputs, processes, and outputs. It also has an environment or setting. Decisions are processes, but, because decisions define flows and paths, it is important to present them separately.

It is often useful to model a system graphically, and there are many ways to do this. We suggest that the standard symbols used for flowcharting (as shown in Figure 2) are used [15]: they have been around for generations, are used across a wide range of disciplines from software engineering to healthcare, and are readily available in “all” popular drawing applications. Lines with arrows represent paths and directions. Ovals represent inputs and outputs. Rectangles represent processes. Processes produce outputs which become inputs to the next process in the pathway, which will be another process, a decision activity, or a final output. Diamonds represent decision processes. Callouts represent comments. The output of a process may be shown as a label on the exiting line. A shaded rectangle that encloses the system represents the environment/setting. The environment provides inputs, and influences processes and outcomes, but these are not explicitly represented. If there is feedback, one or more outputs are connected back to inputs, but this is not shown in the figure. If a care pathway needs to be shown on more than one page, a line to a circle labelled with the page number can be used for navigation.

System diagrams can be much more complex than shown in Figure 2, and can use a bewildering variety of symbols and color mappings. However, it is best when systems diagrams are as simple as possible while still showing all the important factors.

Note that decision trees (which are discussed in the section on health economics below) use a different set of symbols.

## 5. Care Pathway Analysis—A Practical Approach

### 5.1. Recommended Approach to Care Pathway Analysis

It is generally accepted that diligent care pathway analysis should be performed early on in the process of product and evidence development. This is because it allows:The manufacturer and their investors to understand the clinical and commercial potential of the device, develop their market access strategy, build their business plan, and guide marketing.The methodologists or researchers to build health economic models and to design clinical studies to generate the evidence required.Healthcare funders to understand the potential value of the product in clinical and financial terms, and decide whether to support further development.

The Test Evaluation Working Group of the European Federation of Clinical Chemistry and Laboratory Medicine (EFLM TE-WG) have published a practical guide to the development and evaluation of biomarkers which provides a helpful checklist for identifying unmet clinical needs and promotes a consistent language among stakeholders. The tool can be used interactively to explore how the new technology can meet an unmet clinical need [16]. 

However, there are few, if any, published resources on *how* to conduct a care pathway analysis [17]. We suggest that the care pathway analyst considers the following recommendations:Work with the manufacturer to articulate the opportunities for improvement (these will be a source for the statement of value propositions).Work collaboratively with a team that has representatives (direct or proxy) of all stakeholders, including the manufacturer (who will usually act as a proxy for their investors), healthcare payers, healthcare providers (organizations, departments, and professionals), and patients. The EFLM TE-WG tool can be used as a framework for discussion. Patient and public involvement is described in the following section.Review local (i.e., hospital) and national guidelines relevant to the condition to understand the current recommendations on its management and the variation of recommendations across the country.Obtain frequent feedback on documents (narrative and graphic representations of the base case and new care pathways) as they are drafted and redrafted—see Figure 3 and the explanation in the next section. Feedback should also be sought from both experts and patients.oBe aware of the ethical issues and organizational processes around involving patients, the public, and staff in meetings and surveys.oFeedback, advice, information can be obtained through team meetings, individual meetings, focus groups, and surveys.Summarize current evidence on the product, such as analytical validity studies—don’t assume that these have been done or that the results are encouraging.Articulate the clinical scenario: the clinical problem, setting, recommended management.Describe the product, indications for use, strengths and limitations, and how it will be used by whom and where.Describe how the information produced by the product will guide management decisions.Obtain comments on usability and potential utility from stakeholders—they could be patients, clinicians in the ward or clinics, laboratory staff, or methodologists such as health economists.Describe the outcomes (financial and clinical) that are expected.Visit (preferably with an example product) the places where the device and the information produced will be used and talk to potential users of the device and/or the information it provides.oDesk research is necessary. Getting out and seeing the real world and talking to real people is even more necessary.

### 5.2. Example Care Pathway

Figure 3 compares the usual pathway for managing our example ConditionR with the pathway were NewTestRx to be adopted by GPs and family practitioners.

In the usual care pathway, a patient presents in primary care with a history and clinical examination that prompts the physician to consider if ConditionR should be diagnosed. If the answer is Yes, the patient is admitted to hospital where they will either recover or not. If the answer is No, management will be directed by the other conditions that are on the differential diagnosis list. The pathway shows: the setting (primary care); the presenting problem suggests ConditionR should be considered and relevant history taken and examination made; the decision point (shown as the diamond with a label ending with a question mark); the management; and the clinical outcomes.

The proposed pathway for managing ConditionRx when NewTestRx has been adopted is similar to that for usual care, and where processes differ, they are shown in brown: i.e., the use of NewTestRx to rule in/out the diagnosis and to provide online personalized guidance to allow the patient to manage their ConditionR at home.

The figure shows that the clinical outcomes are the same for both pathways. It does not show that the proportions of patients will differ for the corresponding outcomes in the two pathways. For simplicity’s sake it does not show non-clinical outcomes that could be a substantial benefit, such as the rapid time to an accurate diagnosis and improved quality of life when a patient is supported in selfcare. These outcomes would be included in a health economics evaluation. Quantification of outcomes to compare the net benefits of usual and proposed new pathways is discussed in the section on health economics, and shown graphically as a decision tree in Figure 4.

## 6. Patient and Public Involvement and Engagement in Healthcare Research

Patient and public involvement and engagement (PPIE) is becoming a mandatory requirement for healthcare research in the UK. Any major research funder will require a section on PPIE to successfully secure a grant. In February 2019, the Public Involvement Standards Development Partnership which brings together representatives including public contributors from the Chief Scientist Office (Scotland), Health and Care Research Wales, the Public Health Agency (Northern Ireland) and the National Institute for Health Research (England) launched new standards for public involvement. The public involvement standards aim to provide everyone engaged in health research with clear, concise benchmarks for effective public involvement alongside indicators against which improvement can be monitored.

However, the level to which PPIE is incorporated within research projects, or the importance placed on PPIE, will vary. Ultimately, understanding the set of issues that a manufacturer might need to address to support product development, and ultimately marketing, should start with talking to the final market destination, the people that it will help. Understanding the patient’s view of their journey through the healthcare system is critical to the care pathway analysis. 

Patient and public involvement can help to create this care pathway evidence base for the development of the medical device/diagnostic, including answering: Why the device is needed?Will there be a market for it?Would it benefit the patient and/or the healthcare professionals?

Beyond the moral imperative to involve those who will eventually benefit from the medical device/diagnostic, PPIE can improve the quality and success of the research. This includes:Improving the quality of research delivery—public/patient involvement can help develop and review materials that participants will see as part of the research (e.g. patient information sheets, consent forms etc.), to make these more relevant and accessible. This will positively impact on the ease with which participants are able to consider whether to take part and to achieve participant recruitment targets.Providing a different perspective—public/patient involvement can provide the research team with real-life insights and perspectives that are not within the reach of the research team without PPIE. A research team may feel nervous on requesting certain samples for research such as muscle or feces samples, or make assumptions on where the best place or time to hold recruitment will be. Many times we find that people surprise us with their insights or perspectives that either affirm the research approach or offer alternatives which benefit the study’s delivery.

### Putting PPIE into Practice

There are numerous approaches to public involvement and engagement. The medical device/diagnostic under development will often determine what public/patient groups to target and their level of involvement in the project. This may include:Consultation: to ask for views/advice.Collaboration/co-production: researchers and people work together e.g. to identify research questions.User led: people make the decisions about research, e.g. they will be the principal investigator.

There are also many models to involve people in the research and this will depend partly on what aforementioned involvement approach you decide to take. Two of the most common involvement activities are:Setting up a public/patient steering group/advisory board or attending an existing one to consult with them on the research question, study design, feasibility, methodology, recruitment strategies etc.Working with the public/patients who will be co-applicants when applying for funding grants.

Lastly, there are many PPIE professionals within Universities, NHS Trusts and NIHR bodies that can provide support and advice on how to approach PPIE. Often the best place to start on PPIE is a conversation with partners like these, who you can work with to develop the research. You may find this leads to new PPIE opportunities which help to bring the medical device/diagnostic to the market.

## 7. Early Economic Assessment

Early economic analysis is used to estimate the likely cost-effectiveness of alternative testing or treating policies under different circumstances such as the possible results of clinical evaluations of the product, and likely variations in production costs and selling price [5,18,19,20].

Early economic analyses help to identify the parameters which have the most effect on the model’s results, and thus inform decisions about: (i) investing in resources to more accurately determine certain parameters; (ii) the target populations for the product; and (iii) pricing strategies for the product.

Headroom analysis is a type of early economic analysis which compares the decision to develop a product (supply) with the decision to buy it (demand), and provides manufacturers with a value-based price ceiling [21]. It is used early in the development process to help the manufacturer and their investors decide whether or not to proceed (go/no go decision). A headroom analysis can be little more than a “back of the envelope” calculation in the very early stages of product development, or more detailed modelling when the product is close to being ready for production and marketing.

Early economic models are often the basis for full economic studies [22].

## 8. Analytical Validity

### Developing Evidence on Analytical Validity for an IVD

Analytical validity studies for an IVD answer the question: Does the device generate the expected results in the development laboratory, i.e., in a highly controlled environment?

Analytical validity studies provide evidence on a device’s general safety and performance. The performance characteristics that are commonly assessed are summarized in Table 2. The design of these studies varies depending on the device being developed, and there are a number of useful resources which set out these procedures in detail [23,24,25].

Analytical validity studies should be carried out on certified reference material and/or samples which have been assessed using a reference method (the “reference standard”). Sample selection should take into account information from the care pathway analysis such as patient populations, care settings, and common comorbidities. Analytical validity studies should explore and document the effects of common causes of misleading results such as interference by free hemoglobin or bilirubin, variation between types of sample, and the presence of drugs in the sample. 

Although analytical validity is assessed under rigorous laboratory control, it is still desirable to validate with a mind to where the device will eventually be used. In our fictitious example, our early care pathway analysis and economic modelling has shown us that the most promising scenario for NewTestRx is in the GP surgery and patient’s home. Therefore, we must include environmental and logistical conditions which are appropriate for that environment when performing our analytical validity experiments. For example, our test should be proven to work in a wide range of room temperatures, as they are not likely to be tightly controlled in a GP surgery. The skill of the operator is likely to be minimal in this setting, so evidence is needed to show that the test is robust in the face of liquid volume handling or timing variations.

The steps taken in this part of the evidence development pathway principally aim to demonstrate the safety of the device, i.e. that it generates an accurate result under test circumstances. It is, however, prudent to ensure that the methods undertaken in the research laboratory attempt to introduce some elements of ‘real-life’, or else the device is much more likely to fail when we move into our next evaluation stages: clinical validity and utility.

## 9. Clinical Validity

Clinical validity studies address the question: how well does the test perform in the clinical environment? The methods employed include human factors studies [26,27], feasibility studies, and, the focus of this section, diagnostic accuracy studies. These studies are observational by design; this means that the results of the test are evaluated, but not used in clinical practice. The clinicians and patients are often blinded to the results of the test so they cannot change their behavior compared to standard practice. The reason for this design is that there is not enough evidence for the results of the new test to be considered safe to use in practice at this stage. 

### 9.1. Developing Evidence on the Clinical Validity of a Diagnostic Test

Once the analytical validity has been established in the laboratory, evaluation in the clinical setting can be planned using the adaptation of the PICO (Population, Intervention, Comparison, Outcome) framework for clinical interventions [28]. 

PICO adapted for diagnostic accuracy studies is:What is the Aim when using the test?What is the Setting in which the test will be used?For example, home, clinic, hospital.What is the Population?How will participants be recruited: practicalities as well as inclusion and exclusion criteria?What is the Index (new test)?What is the Comparison or reference test?Is a test currently available? If not, what would current practice be to provide an operational definition of making the diagnosis?What Outcome (diagnostic accuracy) measures will be used?Over what length of Time are the outcomes measured?

A small feasibility study is worth the investment in time and other resources to check the feasibility of study design, study management, data collection, and data analysis. It can also be used to assess human and workflow factors that could detract from the usability of the device and its outputs.

### 9.2. Study Designs for Assessing Diagnostic Accuracy

Diagnostic accuracy studies are typically observational cohort studies in which the accuracy of a new test is compared to a reference test in a defined population [19]. However, there are circumstances in which the study design would need to be modified.

An *interventional rather than observational* study design might be appropriate if the results of the new test become available well before the usual test’s results, and prompt action would have a substantial clinical benefit. The study’s ethical review committee would need evidence from analytical validity studies and the scientific literature that the new test’s results can be sufficiently trusted. The ethics committee would also want to know how the comparator test’s results would be used.

A *case-control rather than cohort* study design [29] may be appropriate, particularly in the early stages of evidence development, when the condition being tested for has a low prevalence. For example, in this situation, specimens for the case group might be obtained from a pathology register or sample bank, while specimens for the control group might be obtained from a clinic. However, the risk of biased results with case-control studies must be kept in mind [30].

### 9.3. Studies to be Conducted Alongside Diagnostic Accuracy Studies

The protocol for a diagnostic accuracy study should describe how discrepancies (false positive and false negative results) are to be reported, analyzed, and resolved (if possible). For example, discrepancies could be resolved by repeating the test, ideally on the same sample, and possibly with a different reference standard. The manufacturer of the test will probably also want to investigate the causes for the discrepancies. 

The manufacturer should consider including an assessment of human factors and usability of the test. This could be a simple survey of the test’s users, or a formal study involving observation of users and interviews with people selected on the basis of their exposure to the test and its results.

### 9.4. Measures of Diagnostic Accuracy

Commonly used measures of diagnostic accuracy are:*Counts or proportions of test results*: TP = number of true positive results, e.g. when a patient with ConditionR tests positive; FP = number of false positive results, e.g. when a patient without ConditionR tests positive; TN = number of true negative results, e.g. when a patient without ConditionR tests negative; FN = number of false negative results, e.g. when a patient with ConditionR tests negative.*Sensitivity*: the proportion of people with a condition who test positive = TP/(TP + FN).*Specificity*: the proportion of people who do not have the condition and who test negative = TN/(TN + FP).*Positive predictive value (PPV)*: the proportion of people who test positive and actually have the condition, in the population being tested = TP/(TP + FP).*Negative predictive value (NPV)*: the proportion of people who test negative and do not actually have the condition, in the population being tested = TN/(TN + FN).

### 9.5. Uses and Abuses of Diagnostic Accuracy Statistics

Sensitivity and specificity are useful for comparing the performance of different devices that have been evaluated in different populations. This is because sensitivity and specificity do not change with the prevalence of a condition (provided the same reference standards are used, and the settings and study populations are similar) [31,32,33]. 

Positive and negative predictive values are what patients and clinicians find most useful. However, unlike sensitivity and specificity, predictive values do change *mathematically* with the prevalence of a condition. This has two important consequences:*For clinical use*. A test that is accurate enough to use in a high prevalence scenario may not be accurate enough to be useful in a low prevalence scenario.*For comparing devices*. It is not valid to compare tests on the basis of their predictive values unless the tests have been evaluated on similar populations (i.e., with the same prevalence of the condition) [31,32].

This may seem counterintuitive. So, consider a test with a sensitivity of 90% and specificity of 80% that is used in a high prevalence (50%) population and a low prevalence (5%) population. There are 1000 people in both populations.

In the high prevalence population, the test has positive and negative predictive values of 82% and 89%.

In the low prevalence population, the test has positive and negative predictive values of 19% and 99%.

The calculations are shown in Table 3, and can be checked with the online calculator at https://micncltools.shinyapps.io/ClinicalAccuracyAndUtility/ [34]. The reader is encouraged to further explore the relationship between prevalence and predictive values, for example by using the interactive online app at https://kennis-research.shinyapps.io/Bayes-App/ [35].

### 9.6. The Tradeoff between Sensitivity and Specificity:Ssetting the Threshold for Classifying a Test Result

Ideally a diagnostic test would be both very sensitive and very specific. However, for many tests, this is not possible for technical reasons, and a trade-off has to be made between sensitivity and specificity: the higher the one, the lower the other. The reason for this is that a test provides a measurement, and a threshold value has to be used to decide whether to classify it as high or low. When a high value corresponds to a positive test, the higher the threshold, the fewer tests will be considered positive, and those that are positive are more likely to be true positives. So, the specificity will increase and the sensitivity decrease. When a low value corresponds to a positive test, the higher the threshold value, the more tests will be considered positive, and those that are positive are more likely to be false positives. So, the specificity will decrease, and the sensitivity increase.

The relationship between sensitivity and specificity is usually visualized by graphing (1 – specificity) against sensitivity in a plot called the receiver operating characteristic curve (ROC curve). The relationship between test threshold and sensitivity for a given ROC curve can be explored using the online interactive app at https://kennis-research.shinyapps.io/ROC-Curves/ [36].

ROC curves are useful for two reasons. Firstly, the area under the ROC curve (called the C statistic) is an overall measure of the test’s accuracy. Secondly, by using it together with a range of threshold values, the optimum value (and hence optimum sensitivities and specificities) can be chosen. A simple way of deciding on the threshold value is to select that which makes sensitivity and specificity equal. However, this would not be optimal for diagnosing a condition that is rare, as the low prevalence would mean there would be many false positive results unless the test were highly specific. It would also not be optimal if the test were mainly to be used to exclude a serious condition, as there would be many false negative results unless the test was highly sensitive. In short, to choose the threshold for an optimal balance between sensitivity and specificity requires you to consider the benefits and harms of true and false positive results, the benefits and harms of true and false negative results, and the prevalence in the populations where the test will be used [32,37,38].

### 9.7. Some Tips on Statistical Analysis

1)When planning clinical studies it is good practice (and required by many funders) to estimate the minimum sample size that will allow the desired effect to be observed in the population of interest. The optimal sample size depends, amongst other things, on the acceptable risks of being wrong. For diagnostic accuracy studies, the risk of being wrong might be expressed as a confidence interval around a diagnostic accuracy statistic such as sensitivity. For studies designed to detect a difference in clinical outcomes, the risks of being wrong might be defined by (i) the P-value for accepting a result as statistically significant (the risk of a false positive conclusion), and (ii) the statistical power of the study (the risk of a false negative conclusion). There are applications that can help with the sample size calculations such as G*Power (https://stats.idre.ucla.edu/other/gpower/), which is free and covers a large range of designs (many relevant for clinical utility studies), but is not ideal for diagnostic accuracy studies [39]. JMP (https://www.jmp.com/en_gb/home.html) is more comprehensive but not free [40]. Hajian-Tilaki provides simple explanations and tables to calculate sample size for diagnostic accuracy studies [41]. Choosing the optimal sample size is often difficult and the logistical and financial implications can be large, so it is wise to consult a statistician when planning the study.2)Sometimes the outputs of the test are not binary. Also, test results can be undetermined or inconclusive, for example if they are just above or below the threshold that separate “positives” from “negatives”. Shinkins and co-authors provide guidelines on how test results should be analyzed and reported when they are invalid (result missing or interpretable) or valid (interpretable but near to the threshold) [42]. As a rule of thumb, all results should be reported and, when possible, included in the analysis, for example with sensitivity analyses to assess the impacts of missing, invalid, and borderline results on the outcome.3)When there is no perfectly accurate test to use as the gold standard, an imperfect reference standard must be used. The methods used to take account of the biases introduced by using an imperfect reference standard have recently been reviewed [43].4)Studies reporting accuracy measures, such as sensitivity and specificity, should be replicated in a different population. This is called external validation [44]. External validation is particularly important for studies which select a subset of multiple biomarkers, as the selection results may be significantly different in different populations. Furthermore, estimates of sensitivity and specificity in the sample population where the biomarker selection has been optimized are likely to be optimistic estimates of the test accuracy.5)Sensitivity and specificity may be difficult to translate directly into decisions about patient treatment, for which information presented in the form of probabilities of disease after a positive or a negative test result may be more useful. Two online interactive tools have been developed to clarify the relationship between pre-test (prevalence) and post-test probabilities of disease. Probabilities of disease can be, then, compared with decision thresholds above and below which different treatment decisions may be indicated [33].

### 9.8. Standards for Reporting Diagnostic Aaccuracy Studies

Reports of diagnostic accuracy studies should follow the STARD [45] or TRIPOD [46] guidelines. These are also useful when developing funding proposals, study protocols, and statistical analysis plans. STARD is appropriate where the test result is based on a single measurement. TRIPOD is relevant when the test result is a classification based on several measurements. The FDA guidelines on reporting diagnostic accuracy studies should be consulted [47]. Although they are intended for the USA, they are also useful for products to be marketed elsewhere. 

## 10. Clinical Utility

Clinical utility studies address the question: Can the new test be useful in clinical practice? [5,6,14,48] Utility of diagnostic testing is defined as “the degree to which actual use of the corresponding test in healthcare is associated with changing health outcomes, such as preventing death and restoring or maintaining health” [14]. The range of possible utilities is large, and includes emotional, social, cognitive, and behavioral effects [9,49].

Clinical utility studies are necessary because an accurate test might not provide utility if:It cannot be delivered quickly and to a high-standard in real-life settings (e.g., in complex healthcare settings).Test results do not influence clinical decision-making or patient-outcomes (e.g., the test is sensitive but there is no effective treatment for the condition).

Clinicians, budget holders and policy makers therefore increasingly require evidence of improvement of final outcomes in the patient population, or enhanced quality and efficiency of care or cost-effectiveness, in addition to technical analytical performance. 

### 10.1. Developing Evidence on the Clinical Utility of a Diagnostic Test

Once the clinical validity has been established, evaluation of full pathway outcomes can be planned using the PICO framework [28], adapted for interventional studies of diagnostic tests.

What is the Aim for the test?What is the Setting in which the test will be used?What is the Population?How will participants be recruited: practicalities as well as inclusion and exclusion criteria?What is the Index (new test) and its care pathway?How will the test be delivered; who will be told the results and when; how will this affect subsequent treatment?What is the Comparison or reference test and its care pathway?Is a test currently available? If not, what would current practice be?What Outcome measures will be used to assess clinical utility?In which way might the new test provide advantages over current practice? How will these be measured and analyzed?Over what length of Time are the outcomes measured?

As with clinical validity studies, a small feasibility study is worth the investment in time and other resources to check the feasibility of study design, study management, data collection, and data analysis. 

### 10.2. Study Dsigns for Assessing the Full Pathway Clinical Utility of a Diagnostic Test

The aim of clinical utility studies is to quantify the improvement in health or healthcare system outcomes that the new diagnostic test brings relative to current best practice. To obtain improvements in outcome, the additional information provided by the test should influence clinical decision making.

Clinical utility studies may be *modelling* “what-if” studies, or *interventional* studies, where clinical management is informed by the test’s result. Modelling studies could be performed early in the evidence development pathway, and their results used to inform product development decisions.

A major challenge in clinical utility studies in diagnostics is the indirect way that testing affects patient outcomes, which means it can be difficult to disentangle the benefit of the test from the efficacy of the treatment. Whereas in pharmacological clinical trials the patient outcome is directly related to the treatment, in diagnostic studies the outcome depends on both the test results and the treatment efficacy. In reality, utility studies in diagnostics evaluate a pathway not a single test, so they can be considered as “complex interventions” [50]. It is important to keep complexities such as associated changes in work processes and information flows in mind, not only when planning and reporting the study, but also when designing the strategy for implementation and adoption of a new device [51].

### 10.3. Randomized Controlled Trials

The prototypical design for a clinical utility study of a diagnostic is the full pathway randomized controlled trial (RCT) [14]. Patients enrolled in the study are randomly assigned to one of two or more care pathways. Each pathway is defined by a different testing strategy, and the aim of the study is to compare the eventual outcomes in terms of utilities such as recovery, survival and quality of life.

RCTs that evaluate full pathway outcomes can be impractically large, long running, and expensive. Table 4 outlines some increasingly common trial designs that can improve the efficiency of a RCT. A growing collection of such study designs can be found at http://www.bigted.org/ [52].

### 10.4. Standards for Reporting Clinical Utility Trials

Reports of clinical utility studies of diagnostic tests should follow the CONSORT statement or one of its extensions. These are available at https://www.equator-network.org/reporting-guidelines/consort/ [58].

### 10.5. Studies to Consider Carrying out alongside a Clinical Utility Study

Manufacturers developing a new diagnostic should consider carrying out a diagnostic accuracy study alongside the clinical utility study. This would provide additional evidence on the test’s performance.

Health economic evaluations are often most efficiently conducted alongside a clinical trial, as much of the infrastructure needed to collect data on resource usage and costs will be set up for the clinical trial.

## 11. Health Economic Studies

### 11.1. Developing Evidence on Value for Money and Affordability

Health economics uses decision analysis and a range of study designs to evaluate healthcare services and technologies [20].

### 11.2. Decision Analysis

Decision analysis allows the outcomes from care pathways with different technologies to be compared, for example current and proposed new care pathways that include the use of current and proposed new testing practices.

Decision analysis involves modelling a care pathway by quantifying the inputs and outputs of each process, and quantifying the probabilities of following a particular pathway from a decision node. The variables that are quantified are the parameters for the model.

In health economics, decision analysis typically uses decision trees and Markov models to structure the decision-making process. 

Decision trees and Markov models can be visualized as flowcharts showing the progression of patients through the care pathway. Decision trees show patients progressing from the beginning to the end, without looping back to a previous state. Markov models enable the analysis to take account of a pathway in which a patient can return to a previous state, for example to journey through: become ill, undergo treatment, respond, relapse and return to ill state.

There are a number of computer applications that can visualize and evaluate a decision tree or Markov model, for example (TreeAge [59] and GeNIE modeler [60] for a Bayesian approach, and Microsoft Visio [61]) for visualization of care pathways, workflows and decision analysis. Figure 4 is an example of a simple decision tree, developed in TreeAge, which compares the costs and outcomes of two strategies for diagnosis and management of ConditionR. The square node represents a decision between the current standard of care for patients with suspected ConditionR and a pathway which also incorporates NewTestRx. The ‘payoff’ (triangle node) cost and outcome of each strategy is obtained by multiplying the value of each cost and outcome by its respective probability. These results can be added at the previous chance node (circle node) on a decision tree, in a process known as rolling back the tree. The decision tree shows that a clinical diagnosis is modelled in the same way as diagnosis with a diagnostic test. In other words, a clinical diagnosis can be correct (true positive or true negative) or incorrect (false positive or false negative).

Within a healthcare system, decisions about adopting a medical technology are based on differences in “expected outcomes” between pathways. An expected outcome is the quantified outcome of interest multiplied by the probability of it happening. In the NHS (and many other health systems), the outcomes of interest are costs and effectiveness quantified as utilities. Utilities are net clinical benefits measured as the value a person gives to a particular state. The most commonly used utility in health economics studies is the quality-adjusted life year (QALY), which combines quality of life with length of life [19,20].

Developing a decision model can be a long and expensive process, and is typically not carried out until the final product is available and all evidence of clinical effectiveness has been generated i.e. the uncertainty around effectiveness is low.

Health economics studies inform decisions about adopting a technology to use in the management of a specific clinical problem.

### 11.3. Cost Minimization Analysis

Cost minimization analysis addresses the question, which of two or more clinically equivalent technologies will result in the lowest cost?

### 11.4. Cost-consequences Analysis

A cost-consequences analysis provides disaggregated costs and a range of outcomes. This type of study allows users to use their own judgment about the relative importance of alternative technologies.

Cost-consequences analysis compares the costs from following a care pathway with consequences, i.e. the clinical outcomes. A cost-consequences analysis can help inform decisions that budget holders in service provider organizations must make about the affordability of adopting a product, and thus can support a manufacturer’s adoption planning.

### 11.5. Budget Impact Analysis 

A budget impact analysis provides a measure of the affordability of adopting a medical device by quantifying the effects on the budget of a healthcare provider.

Budget impact analysis goes beyond cost-consequences analysis in that it takes account of the impact on revenues as well as expenditures resulting from the adoption of a new product. The perspective of budget impact analysis is often that of a local or national decision-maker, and can help such decision-makers offset increased expenditure in one budget silo against greater savings in another budget silo.

### 11.6. Cost-effectiveness Analysis

A cost-effectiveness analysis provides a measure of the value of adopting a medical device. It quantifies the impact on costs relative to the impact on net benefits (benefits and harms). The technical name for this is “incremental cost-effectiveness ratio”, or ICER. If the ICER is below the “willingness to pay” threshold (which is between £20,000 and £30,000 for NICE, or higher in priority settings), adoption of the product is considered to provide value for money for the NHS.

The cost-effectiveness plane, shown in Figure 5, is a tool for visualizing the difference that a new product makes to costs and health outcomes, when compared to the base case. It helps us understand the decisions that health economists can support about investing/commissioning/purchasing a product/medical device/clinical test/drug.

The graph shows the difference in effectiveness along the x-axis, and the difference in costs along the y-axis—when comparing the impact of a new product with that from the base case. 

Because there is zero difference in costs and zero difference in effectiveness when the base case is compared with itself, it maps to the origin.

The new product is “mapped” this way to a point in one of the 6 shaded areas in the plane shown in Figure 5. For example, the product could be mapped to area A, B1, B2, C, D1, or D2:A:Products that cost more, but are less effective are mapped to this quadrant. The rational decision about investing in these products is clearly to invest elsewhere.B:Products that cost more, and are more effective need to be assessed for the value that would be provided were they to be adopted. The green diagonal line shows the threshold above which the product would be considered poor value (B1), and below which it would be considered good value (B2).Because there are uncertainties in health economic studies, the threshold line is in reality broader and fuzzier than shown, and thus assessing the value of products near the line requires careful judgement.C:Products that cost less while being more effective make easy investment decisions since they clearly provide better value than the alternative.D:Products that cost less while being less effective require a value judgement about the benefit of the savings relative to the loss in effectiveness. Products below the diagonal green line (D1) may provide an opportunity for savings, while those above the line may not be good value.

### 11.7. Assessing Uncertainty in Health Economics Studies

The aphorism, “All models are wrong, but some are useful” has been attributed to a number of people because it has been for some time a popular way of saying that models should focus on what is important. So, how can we check how wrong our model is in reflecting what is important?

Uncertainties can arise from the structure of the model or from how the parameters are quantified. Figure 2 shows the general form of a system, and can be used as a checklist of potential uncertainties: inputs, processes and their relationships; decision processes and probabilities of their dependent paths; outputs; and the environment/setting. Assessing the sources and importance of uncertainties in health economics models is well formalized and should be done routinely [62,63,64].

The data used in a health economics model can come from new clinical studies, previous studies, data collections (e.g., routinely collected data on service activities and tariffs), and experts (as opinions when primary data is not available). Whatever the source of data, there will be uncertainties in every item. Health economists take account of this inherent uncertainty by exploring in a sensitivity analysis how the outputs of their models change when the data changes. The sensitivity analysis will identify the variables (parameters) which have the most effect on the model’s outcomes.

This analysis provides a formal framework to help decision makers with the following two questions:1)Should the technology be adopted on the basis of current evidence and its uncertainty surrounding clinical and economic outcomes?2)Is further evidence needed in order to support this decision now, and in the future?

Health economic models are a simplification of reality, and to be “fit for purpose” for decision making, they must strike a balance between pragmatism and realism, simplicity and complexity. At early stage modelling, it may be difficult to determine the future care pathway in which a technology could be used and the assumptions to be used when mapping out the pathways for a mathematical model. This is called structural uncertainty. In some instances, particularly at early stages of product development, large structural uncertainties are expected and may not be too concerning. Structural uncertainty can be explored by comparing the results of the model with results from another model which incorporates different assumptions about structure. However, there are also instances when further model development may be required and should be explored at later stages of evidence and product development. For instance, if the model assumptions are at odds with new evidence, guidelines, or changes with policy. In these cases, more detail can be added to a model by combining a decision tree with a Markov model, or by adding a time dependency. Patient level simulations (as opposed to cohort models) more adequately reflect true care pathways and allow simulated individuals to move through the model, rather than propagating the proportions of a cohort. Patient level simulations also allow the accumulating history of the patient to determine transitions within the model. However, the disadvantages of such models are the data requirements. They typically require inputs which are dependent on important patient characteristics, those that determine the natural history of disease. If such datasets are available however, patient level simulations can explore the richness of the data and can be used to explore a range of specific decision problem.

Parameter uncertainty arises from the choice of specific values used to populate the model. The effect the uncertainty has on the model results can be explored through *deterministic* and *probabilistic* sensitivity analyses. The choice of which will be used will come down to the stage in evidence development, and resources available such as time for analysis and the skill and experience of the analyst. 

Scenario analysis allows for the incorporation of other possible future events (such as outbreaks of infectious diseases) or even regional differences in care pathways. This analysis can allow comparison of results in different scenarios and help to provide transferable results and future proof the analysis. 

Deterministic sensitivity analysis allows an exploration of the effect of the uncertainty surrounding each model parameter individually. One by one, each model parameter is varied from its point estimate, or base case value, usually through a predefined, plausible range. The range is often obtained from published information such as a confidence interval. In a univariate sensitivity analysis, each parameter is varied one by one. In a multivariate sensitivity analysis, multiple parameters are varied simultaneously. 

The results of deterministic sensitivity analyses are often presented as a table or a tornado graph. Each row of the table, or bar in the tornado graph, represents a univariate sensitivity analysis for a parameter. The table and tornado graph rank the parameters in order of influence on the outcomes of the model. The results of this analysis provide an understanding on the parameters which drive the outcomes of the model, however it does not take into account correlations between model parameters. This is where probabilistic sensitivity analysis (PSA) should be used [62].

PSA allows the analyst to reflect parameter uncertainty in the outputs of the model [63]. In PSA, this uncertainty is included in the model by choosing statistical distributions for parameter values around the base case value. The distribution chosen will depend on the type of variable and can be defined by the distribution’s key parameters, such as mean and standard deviation for a normal distribution. Further information on the appropriate choice of distribution can be found in [62]. In order to run such models, random number generators are used to draw values for each parameter from their predefined distribution for a single model run. This analysis is repeated a large number of times. For cost-effectiveness analysis (CEA), the results are plotted on the cost-effectiveness plane, allowing the analyst to visualize the distribution of ICERs, as shown in Figure 5. The proportion of the results which fall into the cost-effective region (compared with a cost-effectiveness threshold) gives the overall probability that the intervention will be cost-effective. To compare with multiple cost-effectiveness thresholds, a cost-effectiveness acceptability curve can be plotted. This presents the decision maker with a summary of the uncertainty and can provide an indication of the strength of the evidence in favor of the intervention being cost-effective.

Exploring the uncertainty in a model does not eliminate the potential for a model based estimate of cost-effectiveness to be unbiased, but it makes it more difficult for an analyst to manipulate the analysis directly to provide a point estimate of cost-effectiveness. This is why PSA is required by bodies such as NICE which consider cost-effectiveness evidence in making adoption recommendations [4,48]. 

### 11.8. Value of Information Analysis (VOI)

As we have described, the decision on which to adopt a new technology based on existing evidence will always be uncertain. The decision made on current evidence may be optimal for now, but that doesn’t mean that, once all the uncertainties are resolved, another intervention will not have higher net benefit. Making the wrong decision will have costs in terms of health benefit and wasted resources. Therefore, further evidence may be required in order to support the “right” adoption decision. Value of information (VOI) analysis allows the analyst to formally ask the question: Is further evidence required to support an adoption decision? [62] A key parameter in a VOI analysis is the expected value of perfect information (EVPI). This is the maximum a healthcare system would be willing to pay for perfect information, i.e. a decision with no uncertainty and the maximum the healthcare system would pay for additional evidence to inform this decision in the future. This places an upper bound on future research costs. The expected cost of uncertainty around a decision is determined by the joint probability that the decision made on current evidence will be wrong and the consequences of a wrong decision. The expected cost of uncertainty can be interpreted as the EVPI, since perfect information can eliminate the chance of making a wrong decision. The costs of future research can be compared with the EVPI to determine whether future research would provide value for money or not.

Of course, it is not possible to eliminate all uncertainties associated with health economic modelling. In many instances it is impossible to know all of the possible effects of the model assumptions and parameter uncertainty. But health economic modelling provides a framework for explicitly considering the known uncertainties, and their effect on the model outcomes. As further evidence emerges, the models can be updated to provide more precise estimates. 

### 11.9. Standards for Reporting Health Economics Studies

Authors of reports on health economic studies should use the CHEERS checklist, which is designed to lead to more consistent and transparent reporting, and ultimately, better health decisions [64]. 

## 12. Developing Evidence on Digital Health Devices and Data-driven Technologies

The evidence development pathway for digital health devices and data-driven technologies is similar to that shown in Figure 1.

### 12.1. Evidence Tiers for Digital Health Devices

Classifying digital health technology devices (DHTs), by function (see Table 5), as NICE and the Department of Health and Social Care do, allows them to be stratified into evidence tiers based on the potential risk to users. The evidence level (i.e. series of evaluation studies) needed for each tier is proportionate to the potential risk to users presented by the DHTs in that tier [2].

### 12.2. Code of Conduct for Data-driven Technologies

In addition to the evidence requirements listed in Table 5, the Department of Health and Social Care sets out principles for a code of conduct for data-driven health and care technology [3]: Understand users, their needs and the context.Define the outcome and how the technology will contribute to it.Use data that is in line with appropriate guidelines for the purpose for which it is being used.Be fair, transparent and accountable about what data is being used.Make use of open standards.Be transparent about the limitations of the data used and algorithms deployed.Show what type of algorithm is being developed or deployed, the ethical examination of how the data is used, how its performance will be validated and how it will be integrated into health and care provision. Demonstrate the learning methodology of the algorithm being built. Aim to show in a clear and transparent way how outcomes are validated.Generate evidence of effectiveness for the intended use and value for money.Make security integral to the design.Define the commercial strategy.

### 12.3. Similarities between the Evidence Development Strategies for IVDs and DHTs

The high-level research questions from the evidence tiers for DHTs are similar to those in the evidence development pathway for IVDs. The relationship is shown in Table 6.

## 13. Product Management

Robust product management methods and techniques will help to ensure that evidence development is completed according to scope, timescale and budget. 

*Project planning and management*: know what must be delivered and when, identify where the dependencies are and determine the critical path of the project, which helps to define the shortest timeframe in which the project can be completed. Product management should include appropriate evaluation of potential risks and issues, change and resources management.*Stakeholder strategy*: Identify the stakeholders in the project together with their requirements, planning your communications with them and how they will be involved in the project.*Financial planning*: plan what will be your strategy to fund the evaluation in addition to the development of the new device. Clinical studies especially can be very expensive but there are many funding bodies which can financially support your research. In the UK, the NIHR In Vitro Diagnostic Co-operatives [65] can help you to develop a successful evidence development strategy that can feed into a grant application.*Regulations*: at an early stage in product development, identify the relevant regulators and what evidence is required for marketing approval. In the European Union the new Medical Devices Regulation Medical Devices Regulation (MDR, Council Regulation 2017/745) will be enforced from the 26^th^ May 2020 [66]. For IVDs the date of enforcement is 26^th^ May 2022 (IVDR, Council Regulation 2017/746) [1]. Among other important changes, the MDR and IVDR will required increased evidence on clinical utility. In US the main regulator for market access for medical devices is the Food and Drug Administration, which is also increasing the requirements for evidence on clinical utility [67].*IP protection and publication strategy***:** consider filing patents before sharing information with the wider community through scientific posters and articles. Note that scientific publications are important to obtain funding, buy-in from clinicians, and of course to facilitate adoption in the later stages. So, timing of patents and publications should be carefully planned and pursued.*Prevent waste*: Review and mitigate avoidable sources of waste and failure in the evidence development strategy [68].

The above is not intended to be a comprehensive product management guide, but as a selection of important points to consider.

## 14. Conclusions

We have provided a framework to help manufacturers of IVDs and other medical devices plan and carry out the set of studies that will provide the evidence needed to support product development from initial bright idea through to marketing and adoption. Although we used a hypothetical point of care test as an example to illustrate the ideas, we showed that the framework can readily be applied to other types of IVDs and medical devices, and can be used within healthcare systems other than the NHS in the UK.

The framework is a guide to thinking about evidence development, and is not meant to be used rigidly like an algorithm. Although the graphical representation of the evidence development pathway in Figure 1 may look linear, it is rarely linear in reality. The journey between discovery and implementation is characterized by feedback loops, especially in the more exploratory initial evaluations. At the early stages of the evidence development there is a high risk associated with the invention (e.g., clinical need and fit in the pathway) and with the technology (e.g., possibility that the technology cannot be developed within the constraints given by its clinical application). The funding in the early stages is then limited, and this limits the possible studies that can be delivered to allow appropriate sample collections (adequate sample size in the appropriate patient population). Care pathway analysis can help in this context since it is relatively low cost and can support the identification of the population of interest to allow appropriate early test development. Even when care pathway analysis is carried out, often pilot data are collected in a population that is cheaper to recruit/access in order to obtain funding for more targeted sample collection. The test design and technology are then adapted and refined to maximize its accuracy.

Not all tests require new evidence to be generated for every stage of the pathway. For example, a new diagnostic POCT, as in our hypothetical example, could provide test results faster, cheaper, and/or more accurately than the service laboratory. Evidence development plans would not need to include expensive clinical utility and cost-effectiveness studies. However, it would probably be appropriate to conduct human factors and workflow studies to ensure that there are no usability issues that would block adoption.

The evidence development framework is designed for tests that provide a classification such as condition absent or present. However, tests often provide a measurement on a continuous scale, or may not have a gold standard to use as a comparator for diagnosis. In such cases, it might be better to use the test as a predictor of response to treatment—diagnostic accuracy studies would not be appropriate, but clinical utility studies would be essential.

The framework we suggest includes important issues such as patient and public involvement and engagement that are not well covered in the literature. It thus provides a comprehensive and unique guide to evaluating POCTs, other IVDs and medical devices.

## Figures and Tables

**Figure 1 micromachines-11-00291-f001:**
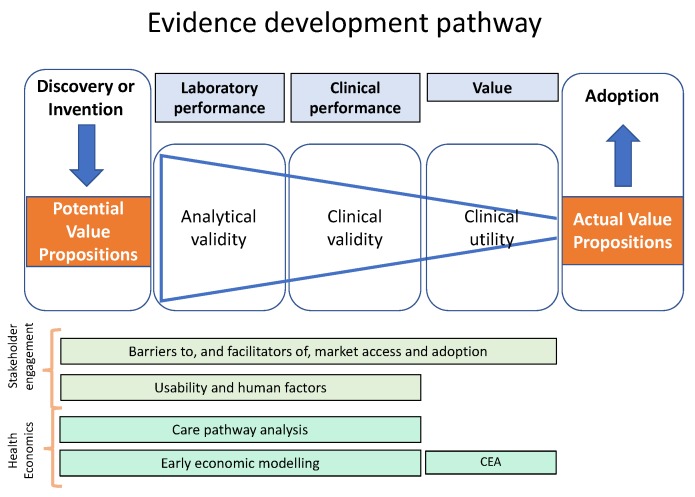
The pathway for developing evidence needed to support the adoption of a new POCT. CEA is Cost-Effectiveness Analysis. The steps are described in more detail in the following sections.

**Figure 2 micromachines-11-00291-f002:**
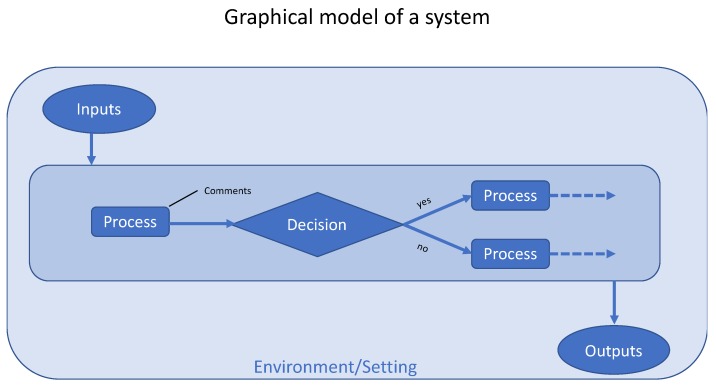
Conventions for symbols in a graphical model of a system.

**Figure 3 micromachines-11-00291-f003:**
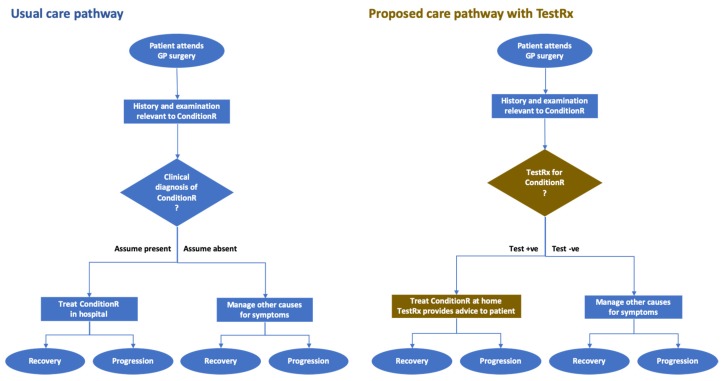
Care pathways for managing ConditionR in current practice and in proposed new practice with the NewTestRx POCT.

**Figure 4 micromachines-11-00291-f004:**
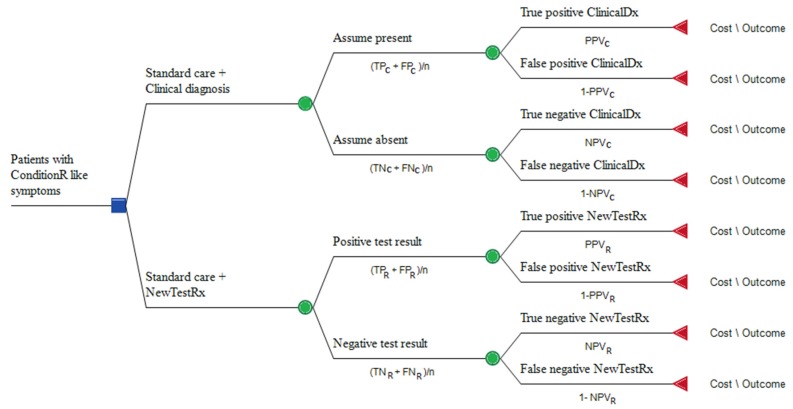
Simple decision tree comparing standard care with standard care plus NewTestRx for ConditionR. TP = number of true positive results, FP = number of false positive results, TN = number of true negative results, FN = number of false negative results, n = total number of patients in that path, PPV = positive predictive value, NPV = negative predictive value. Subscripts (C for clinical diagnosis and R for diagnosis with NewTestRx) identify the relevant branch for TP, FP, FN, TN, PPV, and NPV.

**Figure 5 micromachines-11-00291-f005:**
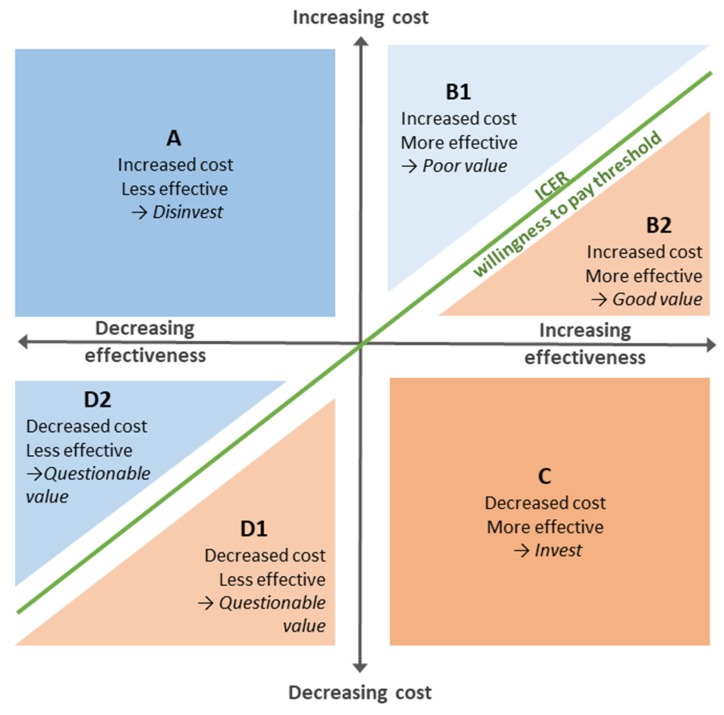
The cost-effectiveness plane with quadrants labelled to show how cost effectiveness study results can inform decision-making.

**Table 1 micromachines-11-00291-t001:** Stages of evidence development to bring a medical device/diagnostic to the market.

Question	Methods to Address the Question
Why is the device needed?	**Articulating the value propositions**^1^ for investors, purchasers, providers, and patients ^2^ **Care pathway analysis** ^1^
Will there be a market for it?	Market research including total available market, serviceable available market, and serviceable obtainable market
Could the product be commercially viable?	Early economic assessment ^1^ to estimate maximum price attainable and identify the information most important for reducing decision uncertainties Barriers to, and facilitators of, adoption, including market access
Does it work in the development lab?	Analytical validity studies ^1^
Can it work in a clinical environment?	**Clinical validity studies**^1^ Human factors analysis and feasibility studies
Would it benefit the patient and/or the healthcare professionals?	**Clinical utility studies**^1^ to establish safety, burdens and benefits to patients, and benefits to the healthcare system
How would it impact the healthcare system?	**Cost-effectiveness analysis**^1^ (value for money) **Cost-consequences analysis** ^1^ (impact of expenditure on clinical outcomes) **Budget impact analysis** ^1^ (affordability within a given budget)
*Will* the product be commercially viable?	Evidence required by the regulator (e.g. equivalence, risks, safety, etc.) Return on investment

^1^ Methods in bold are discussed in more detail in the text. ^2^ The purchaser, provider, and patient may be the same or different depending on the device and health system. For simplicity, we assume that they are all different, as in the UK NHS.

**Table 2 micromachines-11-00291-t002:** Checklist of characteristics typically assessed in analytical validity studies.

Characteristic	Description
Analytical sensitivity	The change in instrument response to changes in analyte concentration.
Analytical specificity	Ability of the method to determine an individual analyte in the presence of interferences.
Trueness	The agreement of the mean of an infinite number of repeated measurements to the reference value—measured with a confidence interval.
Analytical bias	Assessment of the systematic over or under estimation of the reference value i.e. attributable to the method.
Precision	Describes how close the results of repeated measurements are to each other under controlled conditions.
Repeatability	Variability of results when measured in a single sample over a short time period.
Intermediate precision	Variability when repeating the test over multiple days, different operators and using different equipment.
Reproducibility	Variability of results between different laboratories.
Detection limits	The minimum and maximum concentrations of the analyte which can be detected by the test.
Quantification Limits	The minimum and maximum quantities which can be accurately measured by the test.
Linearity	Ability of the test to give results which are directly proportional to the analyte concentration.
Range	The interval over which the test provides results with acceptable accuracy (often dictated by linearity).
Robustness	Ability of the test to withstand small variations in the method parameters e.g. temperature or pH.

**Table 3 micromachines-11-00291-t003:** Predictive values change with prevalence. The tables in the shaded areas are variously referred to as 2x2 tables (when the column and row totals are omitted), confusion matrices, and contingency tables.

**Population with high prevalence**								
population	1000			**Condition**				
prevalence	50%			**Present**		**Absent**		totals
sensitivity	90%	**Test**	**Positive**	**450**		**100**		550
specificity	80%		**Negative**	**50**		**400**		450
+ve predictive value	82%		totals	500		500		1000
-ve predictive value	89%							
**Population with low prevalence**								
population	1000			**Condition**				
prevalence	5%			**Present**	**Absent**		totals	
sensitivity	90%	**Test**	**Positive**	**45**	**190**		235	
specificity	80%		**Negative**	**5**	**760**		765	
+ve predictive value	19%		totals	50	950		1000	
-ve predictive value	99%							

**Table 4 micromachines-11-00291-t004:** Types of randomized controlled trials that can be used to evaluate a diagnostic test. Each of the basic designs has a number of variations.

Study Type	Design of Randomization and Comparison
Randomized controlled trial	Patients are randomized to have either the usual test and its management or the new test and its management.
Marker by treatment designs	Patients are tested, classified into groups such as test positive and test negative, and then randomized to treatment or control arms. The difference between the treatment and control groups in the test positive patients is then compared to the difference between the treatment and control groups in the test negative patients.
Biomarker strategy designs	Patients are randomized to usual care or to care informed by the test results.
Cluster-randomized designs	Study centers with groups of patients rather than individual patients are randomized to the usual or new testing strategy [53].
Stepped wedge designs	The stepped wedge design is an adaptation of the cluster-randomized design in which the new testing strategy is introduced stepwise to a randomly chosen series of clusters over a number of periods of time [54].
Adaptive designs	Adaptive designs allow changes to be made to a trial as information is collected [55,56,57].
Umbrella designs	Several different biomarkers are measured and each determines a different care pathway. This allows several trials to be carried out in parallel.

**Table 5 micromachines-11-00291-t005:** Evidence tiers for DHTs as classified by functional category.

Evidence Tier	Category of Digital Health Technology (DHT) Category, and Required Supporting Evidence *
**Tier 1**	DHTs with potential system benefits but no direct user benefits. ***Evidence of*** Credibility with UK health and social care professionals. Relevance to current care pathways in the UK. Acceptability with users. Equalities considerations. Accurate and reliable measurements (if relevant). Accurate and reliable transmission of data (if relevant).
**Tier 2**	DHTs which help users to understand healthy living and illnesses but are unlikely to have measurable user outcomes. ***Evidence for tier 1, and of*** Reliable information content. Ongoing data collection to show usage of the DHT. Ongoing data collection to show value of the DHT. Quality and safeguarding.
**Tier 3b**	DHTs with measurable user benefits, including tools used for treatment and diagnosis, as well as those influencing clinical management through active monitoring or calculation. It is possible DHTs in this tier will qualify as medical devices.
**Tier 3a**	DHTs for preventing and managing diseases. They may be used alongside treatment and will likely have measurable user benefits. ***Evidence for tiers 2 and 1, and of*** Use of appropriate behavioral change techniques (if relevant). Effectiveness for the intended use (compared to current best practice). Value for money (compared to current best practice). Budget impact Costs and non-monetary consequences Cost-utility (cost-effectiveness)

* Evidence requirements taken from [2], but analogous to those in [3,48].

**Table 6 micromachines-11-00291-t006:** Comparison of evaluation study methodologies for IVDs and DHTs.

Evaluation Question	IVD Methodology	Maps to	DHT Tier
Does the device perform as expected in the development laboratory?	Analytical validity	↔	Tier 1
Does the device perform as expected in the real world?	Clinical validity	↔	Tier 2
Does the device provide value to patients and the healthcare system?	Clinical utility	↔	Tier 3

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
