# Peer review of "How to Ease the Pain of Taking a Diagnostic Point of Care Test to the Market: A Framework for Evidence Development"

_micromachines, 2020, doi:10.3390/mi11030291_

Round 1

Reviewer 1 Report

This 'opinion' article provides the reader with an evaluation guide for the design, development, adoption and manufacturing of a point-of-care device, using a simple model device as a case study. Many of the key aspects in the the development pipeline are covered in-depth, from value proposition and cost-analysis to clinical validity and data management. Importantly, the article highlights important considerations and suggests how to make the best decisions to help deliver a IVD to market.

The article is well-written, easy-to-read and is highly relevant to those in both academia and industry that are developing IVDs.

Author Response

We thank the reviewer for their very helpful comments and agreeing that this is worthwhile topic.

On our review we noted some omissions and errors of cognition which we have corrected.

Reviewer 2 Report

The manuscript provides an exceptionally comprehensive overview of the route taken from point-of-care test conceptualisation to commercialisation. It includes vital links to webpages the reader can consult if they hit a bottleneck as well as suggesting what crucial evidence is needed before progressing. This paper is incredibly helpful and I believe should be consulted by both researchers and manufacturers before considering product commercialisation. Overall this was excellent manuscript and I look forward to it being published so that others can benefit from the imparted information. 

One minor error (although it may be due to misinterpretation) is that in Figure 3 in the current care pathway under failure the left branch says ‘patient progresses’ should that be ‘illness progresses’ or ‘patient regresses’? I’m just not sure what that means in respect to treatment failure? Similarly, on the proposed care pathway if treatment is administered as a result of a positive test result under false positive it says ‘patient recovers’ is that correct? In my mind a patient that is subject to treatment after giving a false positive result will not recover because they have not been treated correctly? I think the true positive and false positive outcomes are the wrong way round in that Figure?

Author Response

We thank the reviewer for their very helpful comments and agreeing that this is worthwhile topic.

We have revised the figure and thank the reviewer for their astute observation.

On our review we noted some omissions and errors of cognition which we have also corrected.